# CT Scan-Guided Fine Needle Aspiration Cytology for Lung Cancer Diagnosis through the COVID-19 Pandemic: What We Have Learned

**Giulia Maria Stella** [1,2,*], **Vittorio Chino** [2,3], **Paola Putignano** [2,3], **Francesco Bertuccio** [2,3], **Francesco Agustoni** [1,4], **Laura Saracino** [2], **Stefano Tomaselli** [2], **Jessica Saddi** [5,6], **Davide Piloni** [2] **and Chandra Bortolotto** [7,8]

1   Department of Internal Medicine and Medical Therapeutics, IRCCS Policlinico San Matteo Foundation, 27100 Pavia, Italy
2   Unit of Respiratory Diseases, Cardio-Thoraco-Vascular Department, IRCCS Policlinico San Matteo Foundation, 27100 Pavia, Italy
3   University of Pavia Medical School, 27100 Pavia, Italy
4   Unit of Oncology, Department of Oncology, IRCCS Policlinico San Matteo Foundation, 27100 Pavia, Italy
5   Unit of Radiation Therapy, Department of Oncology, Clinical-Surgical, IRCCS Policlinico San Matteo Foundation, 27100 Pavia, Italy
6   University of Milano-Bicocca, 20159 Milano, Italy
7   Department of Clinical, Surgical, Diagnostic and Pediatric Sciences, University of Pavia, 27100 Pavia, Italy
8   Unit of Radiology, Department of Diagnostic and Imaging Services, University of Pavia Medical School, 27100 Pavia, Italy
*   Correspondence: g.stella@smatteo.pv.it; Tel.: +39-0382-03369; Fax: +39-0382-502719

**Abstract:** Background and rationale. Novel coronavirus-related disease (COVID-19) has profoundly influenced hospital organization and structures worldwide. In Italy, the Lombardy Region, with almost 17% of the Italian population, rapidly became the most severely affected area since the pandemic beginning. The first and the following COVID-19 surges significantly affected lung cancer diagnosis and subsequent management. Much data have been already published regarding the therapeutic repercussions whereas very few reports have focused on the consequences of the pandemic on diagnostic procedures. Methods. We, here, would like to analyze data of novel lung cancer diagnosis performed in our Institution in Norther Italy where we faced the earliest and largest outbreaks of COVID-19 in Italy. Results. We discuss, in detail, the strategies developed to perform biopsies and the safe pathways created in emergency settings to protect lung cancer patients in subsequent therapeutic phases. Quite unexpectedly, no significant differences emerged between cases enrolled during the pandemic and those before, and the two populations were homogeneous considering the composition and diagnostic and complication rates. Conclusions. By pointing out the role of multidisciplinarity in emergency contexts, these data will be of help in the future for designing tailored strategies to manage lung cancer in a real-life setting.

**Keywords:** *Sars-Cov2*; lung cancer; biopsy

## 1. Introduction

Since being first detected in Wuhan (China) in 2019, *Sars-Cov2* infection rapidly spread worldwide, with Italy being one of the first and most severely involved countries. On 11 March 2020, the World Health Organization (WHO) officially declared COVID-19 a pandemic (WHO website at https://www.who.int/director-general/speeches/detail/who-director-general-s-opening-remarks-at-the-media-briefing-on-covid-19---11-march-2020; date of last access: 25 January 2022). Since February 2020, more than 19,000,000 cases and a case fatality rate of 0.2% (Istituto Superiore di Sanità. Website at: https://www.epicentro.iss.it/en/coronavirus/sars-cov-2-dashboard, date of last access: 25 January 2022) were reported in Italy, with higher mortality rates during the first pandemic wave. The sudden

outbreak of the pandemic forced healthcare systems to rapidly employ economic and human resources for the diagnosis and treatment of COVID-19 patients. Restrictive measures adopted by governments, the reallocation of limited resources to face the emergency, reduced access to hospitals, and progressive exhaustion of infrastructures had a negative impact on the diagnosis of many diseases, resulting in the disruption of healthcare services. Among others, lung cancer care pathways were heavily affected [1]. Over 2020 and 2021, several studies worldwide pointed out the effects of the first pandemic waves on the diagnosis and treatment of lung cancer patients, leading to reduced access to screening programs [2] and increased delays in diagnostic and therapeutic procedures [3–5]. Therefore, a more advanced stage at diagnosis was observed [6], with limited therapeutic options and worse prognosis. The investigation of the impact of the diagnostic delay on the survival outcomes in the United Kingdom estimated that additional 1235–1372 deaths could be expected [7]. On the other hand, significant increase in the incidental diagnosis of lung cancer was reported during COVID-19 management [8]. In this complex landscape, continuous efforts to face the emergency resulted in progressive reorganization of Healthcare Systems by creating safe and specific diagnostic paths for lung cancer patients. In this retrospective observational study, we analyzed the impact of COVID-19 from 2020 to the present day (June 2022) on the diagnostic workup of lung cancer patients referred to our Institution and compared the results to pre-pandemic situations. We evaluated the variations in the number of patients, characteristics, diagnostic approach, histotypes, incidental diagnoses rate, stage at diagnosis, and possible changes following the introduction of vaccines and reorganization of the healthcare system.

## 2. Materials and Methods

A consecutive series of 276 patients were retrospectively evaluated in the present study, all presenting at evaluation for peripheral lung masses suspected to be cancer. All the patients, after providing informed consent, underwent CT scan-guided transthoracic fine needle aspiration (FNA) biopsy of the pulmonary lesions according to an already described procedure [9–11]. The interval times of the study were from March to August 2019 (pre-COVID-19 pandemic) and a corresponding interval time from April to September 2020 (during and immediately after the COVID-19 pandemic). The study entered a main project that was approved by the local Ethical Commission, and each enrolled patient provided written informed consent before enrollment (Comitato di Bioetica, Fondazione IRCCS Policlinico San Matteo, approval numbers: protocol #20090002344; procedure #20090019080; date of approval: 3 June 2009). Detailed data of the enrolled patients are available in Table 1. No specific inclusion criteria were applied: this study included all patients who came under our observation before and immediately during the pandemic and for whom an indication has been given to carry out the procedure after multidisciplinary evaluation regardless of comorbidity, age, ethnicity, and gender. The two cohorts, defined as pre-COVID-19 and COVID-19, were homogeneous in terms of demographic and clinical features (Table 1) and differ in observation time only. For each patient, the following issues were evaluated: the lesion pattern (solid, partly solid, or non-solid), its dimension, the complications during the procedure, as well as the adequacy of the sampling and the time required for diagnostic confirmation. The procedure was performed with diagnostic intent in the presence of a lesion detectable on CT scan; in the vast majority of cases, PET scan was not available at the study time. Based on the histology report and any subsequent or follow-up imaging procedures, each examination was evaluated as true positive, true negative, false negative, inconclusive, or inadequate. The result was considered inadequate if the pathologist described an insufficient number—or even the absence—of cellular elements. A case was considered as false negative if the negative cytology on FNA-CT-guided biopsy was not coherent with subsequent follow-up imaging or further invasive procedures.

**Table 1.** Details of patients, procedures, and lung lesion characteristic in the two cohorts analyzed (pre-COVID-19 and COVID-19). When available, corresponding % are indicated. ADC: adenocarcinoma, SCC: squamous cell carcinoma, undiff: undifferentiated.

| | Pre-COVID-19 Cohort | COVID-19 Cohort | *p*-Value |
|---|---|---|---|
| Total scheduled procedures | 135 | 141 | |
| Procedures carried out | 109 (80.74%) | 120 (85.1%) | |
| Females | 42 (38.5%) | 41 (34.1%) | |
| Males | 67 (61.5%) | 79 (65) | |
| Smoking habit (current/past) | 101 (75.5%) | 107 (76%) | 0.001 |
| Average age (yrs) | 70.32 | 69.25 | |
| Procedures cancelled (% of scheduled) | 26 (19.26) | 21 (14.89) | 0.001 |
| Nodule resolution | 7 (26.92%) (2 GGO-5 solid) | 10 (47.6%) (3 GG0-7 solid) | 0.59 |
| Technical/Organization problems | 8 (30.76) | 6 (28.57) | |
| Health problems | 3 (11.53%) | 2 (9.5%) | |
| Poor patient cooperation | 6 (23.07%) | 1(4.7%) | |
| *Sars-Cov2* infection | -- | 1 (4.7%) | |
| Not known | 2 (7.69%) | 1 (4.7%) | |
| Average nodule size (mm) | 30.219 | 32.3 | |
| Complications | 27 (24.77%) | 17 (14.1%) | 0.02 |
| Pneumothorax | 17 (62.29%) | 14 (82.35%) | 0.05 |
| Hemothorax/Hemoptoe | 10 (37.7) | 3 (17.64%) | 0.05 |
| Hospitalization | 13 (48%) | 0 | |
| Average days of resolution | 6.33 | 4.11 | |
| Hospitalized pts | 8.28 | 0 | |
| Non-diagnostic procedures (% of performed) | 10 (9.17) | 7 (5.8) | 0.59 |
| Repeated procedures | 5 (50%) | 0 | |
| Diagnostic confirmation on repeated procedures | 5 | 0 | |
| Days between the first visit and procedure | 23.25 | 19.95 | |
| Days between diagnosis and start of treatment | 32.75 | 39.47 | |
| Nodule pattern | | | |
| GGO | | | |
| Total number | 7 (4.58%) | 10 (8.33) | 0.59 |
| Spontaneous resolution | 2 (38.57%) | 3 (30%) | |
| Biopsies | 5 | 7 | |
| ADC | 4 (80%) | 6 (85%) | |
| Non-diagnostic | 1 (20%) | 1 (15%) | |
| MIXED | | | |
| Total number | 3 (2.75%) | 2 (1.6%) | 0.47 |
| Spontaneous resolution | 0 | 0 | |
| Biopsies | 3 | 2 (1.6%) | |

**Table 1.** *Cont.*

|  | Pre-COVID-19 Cohort | COVID-19 Cohort | *p*-Value |
|---|---|---|---|
| ADC | 2 | 2 (1.6%) | |
| Metastatic lesion | 1 (ovary) | 0 | |
| Non diagnostic | 1 | 0 | |
| SOLID | | | |
| Total number | 101 | 114 | 0.56 |
| % planned procedures | 74.81 | 80.85 | |
| Biopsies | 97 (88.9%) | 111 (78.72) | |
| ADC | 37 (38.14%) | 53 (47.7%) | |
| EGFR activating mutations (% of ADCs) | 11.5 | 12.7 | |
| ALK rearrangement (% of ADCs) | 2 | 2 | |
| SCC | 13 (13.40%) | 17 (15.3%) | |
| NSCLC undiff. | 3 (3.09%) | 9 (8.1) | |
| PDL1 TPS 5–50% (% of all NSCLCs) | 32.5 | 35.6 | |
| Metastatic lesions | 8 (8.24%) | 3 (2.7%) | |
| SCLC | 3 (3.09%) | 2 (1.8%) | |
| Carcinoid | 2 (2.06%) | 2 (1.8%) | |
| Chondromas | 2 (2.06%) | 1 | |
| Mycobacteria | 2 (2.06%) | 0 | |
| Inflammation | 1 | 5 (0.5%) | |
| Fungal infection | 0 | 2 | |
| Sarcoidosis | 1 | 0 | |
| Hamartoma | 1 | 0 | |
| Abscess | 1 | 0 | |
| Myxoid chondroma | 1 | 0 | |
| Non diagnostic for cancer | 8 (8.24) | 9 (8.1%) | |
| Non diagnostic | 9 (9.27) | 7 (6.3%) | |

Qualitative variables will be then analyzed and described as counts and percentages of each category. Comparison with the two cohorts will be performed. By setting two-sided type I error at 7% and confidence level at 95% with this sample size, we will be able to detect the differences in the observed frequencies to be significantly higher if the sample size is between 97 and 115 cases. The sample size analyzed is adequate for the study goal.

The goal of the study was to define a procedure that evaluated whether the inhomogeneity of the observed frequencies in the two cohorts was due to random sampling rather than to the effect of COVID-19. To check if the observed frequencies in the COVID-19 cohort match the expected frequencies (pre-COVID-19 cohort), the hypothesis test $\chi^2$ has been performed. In the presence of an expected number of individuals less than 5, the Fisher's exact test was applied. Statistical analysis on quantitative variables was not performed since it is not coherent with the scope of this work.

### 3. Results

Overall, our study included a total of 276 consecutive cases of single lung nodules/masses that were evaluated in our Institution. Of them, 229 cases underwent transthoracic CT-guided biopsy with diagnostic intent based of the suspicion of lung cancer. A total of 26 and 21 procedures were cancelled in the pre-COVID-19 and COVID-19 period,

respectively; the main causes were related to spontaneous nodule resolution, poor patient collaboration to the maneuvers, patient's poor clinical conditions, and CT scanner malfunctioning (causing the need to reschedule the procedure); only one case out of the 21 was cancelled due to *Sars-Cov2* concomitant infection. Two different cohorts were identified: (i) 109 patients who came to our observation in the pre-COVID 19 pandemic in a six-month interval; (ii) 120 patients who were evaluated during the COVID-19 and immediate post-emergence time. The goal of our work was to evaluate the impact of the pandemic on the diagnostic work-up in case of suspected lung tumors. Details of each analyzed subgroup are represented in Table 1. Pre-pandemic performances of FNA CT-guided biopsy procedures were coherent with literature data [12–14]. In our experience, patients with suspected lung cancer were generally first sent to pulmonologists, and less frequently to oncologists and thoracic surgeons. Then, the first multidisciplinary discussion regarding the cases was performed to decide which diagnostic approach would be more suitable, based on imaging data, clinical history, and the performances status of each patient. The pneumology division was also involved in the management of biopsy maneuver complications. The obtained samples were processed as cell-blocks in formalin, similar to what was performed before the pandemic [10]. The quantity of neoplastic material obtained through this approach was sufficient for genotyping (Table 1) and was overall sufficient for planning the treatment for all the patients according to standard guidelines. Based on the pathologic results, if inconclusive or inadequate, the case was revised by all specialists and a different biopsy approach could be chosen if available. Once cyto-histologic diagnosis was reached, the multidisciplinary team defined the treatment strategy based on a complete personalized perspective. The interval between the first entrance into the diagnostic and therapeutic path and the treatment start was about 30 days. During the pandemic, even though pneumologists, anesthetists, and thoracic surgeons were mostly devoted to COVID-19, the multidisciplinary management of lung cancer patients resulted in a prompt switch of diagnostic procedures with no significant delays in diagnosis confirmation and therapeutic decisions. The multidisciplinary work-up and its adaptation in COVID-19 pandemic are represented in Figure 1. Our hospital is divided into independent pavilions, which naturally allows to create safe pathways for COVID-free patients (external open-hair corridors). We performed a naso-pharyngeal swab the day before the procedure and administered a health questionnaire the day of the procedure before admitting the patient to the radiology department. Our safety protocol was similar to those already reported in literature for other purpose and produced the same results: the rate of collateral diagnosis of suspected COVID infection during the control scan of the procedure was very low (only one patient) and comparable to those reported (0.2%) [15,16]. It is interesting that the rate of procedures cancelled or postponed (14%) was significantly lower than that reported for lung cancer screening (32.7%) probably due to the different nature of the examinations (voluntary nature of the screening compared to the potentially disruptive results of the diagnostic biopsy procedure). Within respect to the fine needle aspiration procedures, for both lungs and other organs, during COVID-19, there was a substantial difference between the performance rates obtained in our Institution and the most frequently published pandemic data that reported a substantial delay and/or a decrease in the number of cases [17–19]. Some modifications of pre-existing processes, among which the implementation of room ventilation, personal protective equipment and face masks, have been applied to allow the continuation of healthcare professionals' work in the safest settings [20–22]; in some cases, FNAB efficiently replaced core biopsies [23]. To evaluate whether the inhomogeneity of the observed frequencies was due to causal sampling rather than to the presence of the emergency (COVID-19), we applied the statistical test $\chi^2$ test. In this case, as the number of degrees of freedom was equal to one for each single data analyzed, we also applied the Yates correction (or the continuity correction test), thereby obtaining values compatible with the hypothesis that there was no effect due to COVID-19, and that the data observed belonged to the same distribution of those expected. In other words, the statistical analysis demonstrated that the two cohorts analyzed (pre-COVID-19 and COVID-19) were

indistinguishable (no significant differences) in terms of the diagnostic rates and major complications. Indeed, based on the p-value observed for the most relevant qualitative variable, data from Table 1 are compatible with the hypothesis that the two cohorts do not differ due to the effect of COVID-19.

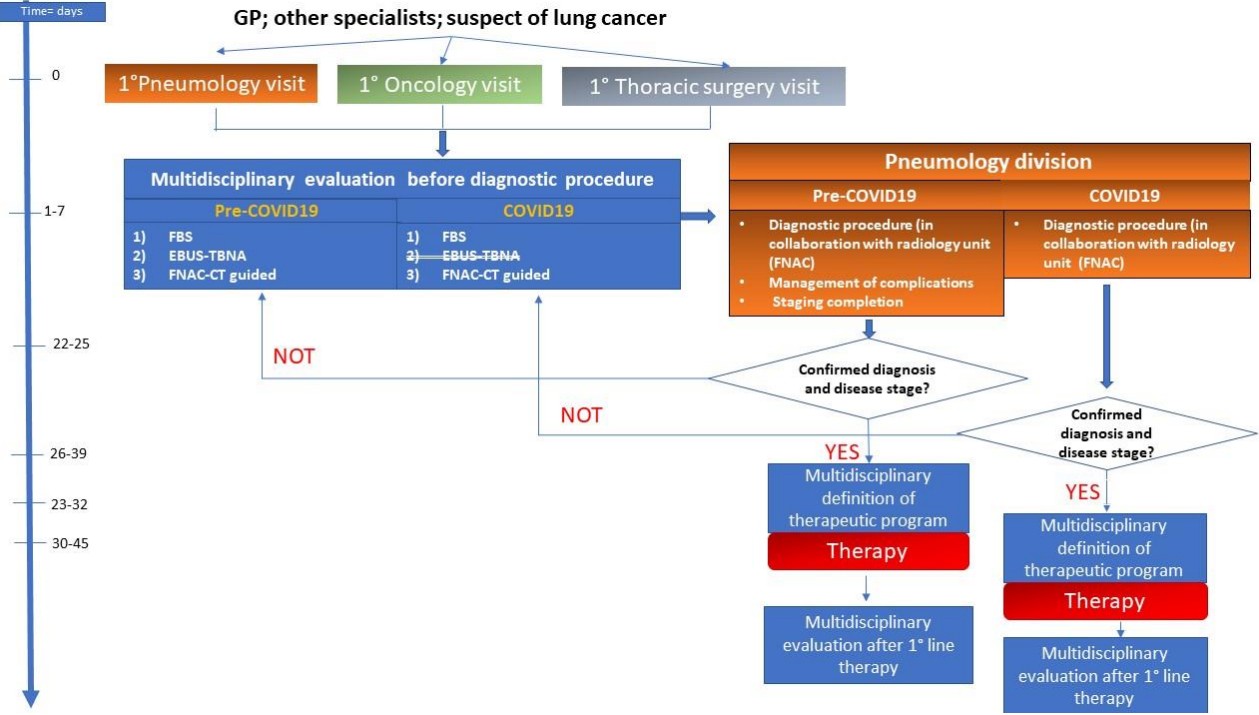

**Figure 1.** Multidisciplinary management of lung cancer diagnosis and treatment before (Pre-COVID-19) and during the first COVID-19 wave.

## 4. Discussion

If compared to the published data, results of the present study showed some unexpected findings. The first is that the performance rates before and during the first wave of the pandemic were almost comparable. This result was reached through an efficient and immediate reassessment of the multidisciplinary committee taking charge of these patients, as represented in Figure 1. It should be underlined that, during the COVID-19 pandemic, pulmonologists and thoracic surgeons were mostly moved to COVID-19 wards. This point importantly limited the access of patients for diagnostic and therapeutic procedures. In particular, the activity of service for endobronchial ultrasound-guided transbronchial needle aspiration (EBUS-TBNA) was significantly reduced for diagnostic purposes, and only nine examinations were carried out exclusively on outpatient subjects in the interval from March to June 2020. Moreover, in our Institution, radial probes were not available in both pre-COVID-19 and during the pandemic. The unavailability of anesthetists imposed the conduction of each procedure under a mild sedation regimen, with the maintenance of consciousness and the use of topical local anesthesia. The staging phases of lung cancer management were assured by performing PET examinations, planned after the multidisciplinary discussion of each case. EBUS-TBNA was conducted as an outpatient service, in a dedicated COVID-19-free area of the Hospital, and the procedures were carried on in the absence of a dedicated anesthetist (data regarding EBUS performances and management during the pandemic goes beyond the scope of this paper and will be discussed in detail in a dedicated work).

This led to the fact that, during the pandemic, most pulmonary masses that could have been identified via endoscopic diagnosis were identified with FNA imaging-guided biopsy. In detail, 23 patients had a pulmonary lesion with a diameter of 50 mm or more.

However, it should be underlined that early lung cancer diagnosis was not delayed as the median dimension of the nodules was about 31 mm.

Overall, the study shows the difficulties in managing non-*SARS-CoV2* disease in an emergency situation, with an increased risk of cancer mortality; the results suggest the need to change behaviors and algorithms as well as make few but significant modifications in a fully clinical multidisciplinary practice. During the COVID-19 pandemic, multidisciplinary board meetings were reduced or conducted online. Literature data agree in underlying that the activity of thoracic endoscopy and surgery services was significantly impaired worldwide by the pandemic [24]. Notably, multidisciplinary meetings were completely cancelled in a percentage varying from 20 to 60% and this was reflected in the delay in lung cancer diagnosis, in outpatients visit, changes in treatment plans, and in a drop of clinical and surgical activities [3,25–29]. The comparison between data from the two populations evaluated allows some relevant considerations. The first is that, even in the first COVID-19 pandemic, lung cancer diagnosis was not delayed, although important remodulation of patient management was required. This data strongly indicate the value of the multidisciplinary group not only in terms of the clinical advantage but also for its intrinsic fluency and dynamics, which were revealed as key issues in case of emergencies. During the pandemic, two pulmonologists were dedicated to the management of lung cancer (or suspected lung cancer) patients in a COVID-19-free area of the Clinic. This approach that was decided in an emergency context at the pandemic outbreak allows us to continue diagnostic and therapeutic activities with very similar pre-COVID-19 number of patients. The second point is that, based on the pandemic experience, a structured "microbial-free" path would be designed in hub hospitals with the aim to preserve diagnosis and treatment of frail and immunodeficient patients. Another issue was that the rate of procedural complications in the COVID-19-cohort was, quite surprisingly, significantly lower than the expected rate. The reasonable explanation is that such an emergency raised operator awareness that resulted in safer procedures. Another point was the more time required for performing the procedure during the pandemic.

There are several limitations of this study, including the following: (i) it is a retrospective study with consequent probability of selection and reporting biases; to avoid this kind of bias, we focused on the comparison between the two cohorts analyses and showed that no statistically significant differences emerge between the two populations; (ii) there was a short follow-up interval so we could not analyze long-term results and compare them between the two cohorts; and (iii) we have limited data regarding the history of any lung cancer in the past for the patients.

## 5. Conclusions

Within respect to the biopsy techniques, the results of our study unveil some unexpected implications. It should be noted that, although many instructions have been published to suggest procedures to protect patients and operators in case of biopsy techniques [30,31], a significant decrease has been reported in the number of FNA procedures with ROSE and EBUS-FNA [32]. In parallel, the role of liquid biopsy was identified [22]. The findings of the present work were similar to the information in the published literature regarding the bronchoscopy procedures. On the other hand, quite surprisingly, the flow of patients who underwent transthoracic CT-guided FNA biopsy did not decrease. Notably, no significant differences emerged between cases enrolled during the pandemic and those before and the two populations were homogeneous considering the composition and the diagnostic and complication rates. It can be concluded that this kind of approach is safe and efficient in allowing the diagnosis of lung cancer, and in particular settings, it could replace/substitute flexible bronchoscopy. Moreover, these relevant data indicate the clinical relevance of the early management of suspected lung cancer by the multidisciplinary teams to assure not only tailored diagnostic and treatment plans but also flexibility and skill that were indispensable in unexpected and emergency situations.

**Author Contributions:** Conceptualization, G.M.S., C.B., V.C., P.P. and F.B.; data collection: V.C., P.P., F.B., D.P., J.S., F.A., L.S., S.T., G.M.S. and C.B.; validation, G.M.S.; data curation, G.M.S., D.P. and C.B.; original draft preparation, G.M.S., D.P. and C.B.; writing—review and editing, G.M.S., D.P. and C.B. All authors have read and agreed to the published version of the manuscript.

**Funding:** Ricerca corrente 5x1000-2020 (cod. 090000X121–progetto 08050122) to G.M.S.

**Institutional Review Board Statement:** The study was conducted in accordance with the Declaration of Helsinki, and approved by the Ethics Committee of Comitato di Bioetica (protocol code #20090002344 and date of approval 3 June 2009), and Fondazione IRCCS Policlinico San Matteo (protocol code # 20090019080 and date of approval 3 June 2009).

**Informed Consent Statement:** Informed consent was obtained from all subjects involved in the study.

**Acknowledgments:** G.M.S. would like to thank Andrea Marchelli for the support in statistical analysis of data and Elena Morganti and Benedetta Marchelli for continuous encouragement.

**Conflicts of Interest:** The authors declare no conflict of interest.

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
