# Peer review of "CT Scan-Guided Fine Needle Aspiration Cytology for Lung Cancer Diagnosis through the COVID-19 Pandemic: What We Have Learned"

_tomography, doi:10.3390/tomography9020061_

Round 1

Reviewer 1 Report

Introduction:

Line 37:  The WHO website URL must be included in the reference list at the end (mentioning the date of access) and not in the main text. Same for the lines 40-41

Material-methods: There is no ethics committee approval, please provide

There is no discussion section and therefore the conclusions section is too long

Author Response

We really thank the Reviewer for comments and for suggestions which improve the quality of the manuscript. We have revised and restructured the manuscript. Below the point-by-point answers (A).

C.1 Line 37:  The WHO website URL must be included in the reference list at the end (mentioning the date of access) and not in the main text. Same for the lines 40-41

A1. We thank the Reviewer for this comment and the website info ahs been added as references

C2. Material-methods: There is no ethics committee approval, please provide

A2. We thank the Reviewer for pointing out this criticism; a dedicated referencein the method section  has been added as follows: The study entered a main project that was approved by the local Ethical Commission, and each enrolled patient gave written informed consent before enrollment (Comitato di Bioetica, Fondazione IRCCS Policlinico San Matteo, approval numbers: protocol #20090002344; procedure # 20090019080; date of approval: 3 June 2009

C3. There is no discussion section and therefore the conclusions section is too long

A3. We thank the Reviewer for suggesting this change. The discussion wasn’t defined in the submitted version of the text since it was a Communication. However we agree with this comment and the text has been divided into discussion and conclusion sections, respectively

Reviewer 2 Report

Thank you for the opportunity to review this paper
This article reported the monocentric experience of strategies developed to perform pulmonary biopsies during the COVID-19 pandemic.
the authors described the strategies developed to ensure an adequate treatment path for patients with lung suspicious lung nodules
However, the topic of this study may not arouse particular interest in the scientific community

Author Response

We really thank the Reviewer for careful revision of our manuscript. Although according to the Reviewer, the topic of this study may not arouse particular interest in the scientific community. However the scope of the manuscript is to describe the strategies developed to ensure an adequate treatment path for patients with lung suspicious lung nodules during the early pandemic. The novelty and interest of this study is related to possible exploitation of its results in the design of dedicated programmes in case of future emergency. Indeed, the main message regards the relevance - in this perspective -  of multidisciplinary teams to assure, tailored diagnostic and treatment plans, but also flexibility and skill that revealed as indispensable in front of unexpected and urgent situations.

Reviewer 3 Report

It was overall a good write up. However, I do have some suggestions, if you feel appropriate, please make those changes.

Firstly is the study retrospective? you started offsaying it is then it sounded like you tried to recruit patients to your study.

I dont know, how valid the results are? The case volumes decreased tremendously during covid and so did the procedures. Most everyone had virtual platform to discuss cancer work up or cancer patients.

There is no inclusion or exclusion criteria, and no indication that there was no bias in recruiting patients.

DO you think that because of the biopsy technique, the diagnosis has improved? I think the patients who did not get worked up got diagnosed? What if patients underwent trans-thoracic biopsy?

No limitations to your study written up.

DO you think decreased case volume during covid impacted decreased procedure complication rate? There was more time to do the procedure?

Also, could these nodules be from prior covid infection or bacterial pneumonia? For non-diagnostic samples, were the repeat biopsies positive for cancer? 

Were the patients recruited have any smoking history or history of any lung cancer in the past?There can be so many confoudning factors that can bias the study.

English can be improved and also fix some spelling errors and grammatical errors.

Author Response

It was overall a good write up. However, I do have some suggestions, if you feel appropriate, please make those changes.

We really thank the Reviewer for careful reading of the manuscript and the constructive remarks. We have deeply revised the structure of the paper and the reference section. Below the point-by-point answers (A).

C1. Firstly is the study retrospective? you started offsaying it is then it sounded like you tried to recruit patients to your study.

A1. We thank the Reviewer for pointing out this point and the retrospective nature of the study has been defined in the method section.

C2. I dont know, how valid the results are? The case volumes decreased tremendously during covid and so did the procedures. Most everyone had virtual platform to discuss cancer work up or cancer patients.

A2. We thank the Reviewer for pointing out this comment. The results of the study are valid according to their statistical significance. By setting two-sided type I error at 7% and confidence level 95%, with this sample size we will be able to detect as significantly higher the differences in the observed frequencies the sample size should be comprised between 97 and 115 cases. The sample size analysed is adequate for the study goal. It should be noted that the goal of the study was’ t that of creating a new database of cancer diagnostic work-up but to report that the multidisciplinary strategy that was defined during the pandemics allowed performance rates ( in terms of lung cancer diagnosis) similar to those in standard (pre-COVID19) conditions.

C3. There is no inclusion or exclusion criteria, and no indication that there was no bias in recruiting patients.

A3. We thank the Reviewer for this comment. This study concerned all patients for whom an indication has been given to carry out the procedure after multidisciplinary evaluation regardless of comorbidity, age, ethnicity, and gender. To avoid a selection bias we focused on the comparison between the two cohorts analyses and showed that no statistically significant differences emerge between the two populations.

C4. DO you think that because of the biopsy technique, the diagnosis has improved? I think the patients who did not get worked up got diagnosed? What if patients underwent trans-thoracic biopsy?

A4. We thank the Reviewer for this comment. Data analysis showed that no differences emerge in diagnostic rates between the pre COVID-19 and COVID-19 cohorts.

C5. No limitations to your study written up.

A5. We thank the Reviewer for this suggestion. The text has been implanted as follows, also coherently with comment 8. There are several limitations in this study, including the following: (i) it is a retrospective study with consequent probability of selection and reporting biases; to avoid this kind of bias we focused on the comparison between the two cohorts analyses and showed that no statistically significant differences emerge between the two populations; (ii) there was a short follow-up interval so that we cannot analyze long time results and compare them between the two cohorts; (iii) we have limited data regarding the history of any lung cancer in the past for the patients.

C6. DO you think decreased case volume during covid impacted decreased procedure complication rate? There was more time to do the procedure?

A6. We thank the Reviewer for this suggestion. The differences in complication rates wasn’t statistically significant; this unexpected result should in part related to the more time to do the procedure during the pandemic. The text has been implemented accordingly

C7. Also, could these nodules be from prior covid infection or bacterial pneumonia? For non-diagnostic samples, were the repeat biopsies positive for cancer? 

A7. We thank the Reviewer for this comment. Although the number are too small to perform statistical analysis (this is why this point is not included in the text), the biopsies were repeated after three months. In one case the diagnosis of cancer was reached.

C8. Were the patients recruited have any smoking history or history of any lung cancer in the past?There can be so many confoudning factors that can bias the study.

A8. We thank the Reviewer for pointing out this criticism. It was discussed in the limitation paragraph as above suggested (C5). Data regarding smoking history has been added in the table.

C9. English can be improved and also fix some spelling errors and grammatical errors.

A9. We thank the Reviewer for careful revision of the text. Typo and grammatical errors has been revised

Round 2

Reviewer 2 Report

The modifications introduced by the Authors have made the paper more complete
and exhaustive.
The results obtained is very interesting, supporting the validity of the work
reorganization adopted during the covid pandemic